# A Coal Mine Tunnel Deformation Detection Method Using Point Cloud Data

**DOI:** 10.3390/s24072299

**Published:** 2024-04-04

**Authors:** Jitong Kang, Mei Li, Shanjun Mao, Yingbo Fan, Zheng Wu, Ben Li

**Affiliations:** Institute of Remote Sensing and Geographic Information System, Peking University, Beijing 100871, China; 2101210061@stu.pku.edu.cn (J.K.); sjmao@pku.edu.cn (S.M.); ybfan@stu.pku.edu.cn (Y.F.); zheng_wu@pku.edu.cn (Z.W.); benli@pku.edu.cn (B.L.)

**Keywords:** point cloud, underground tunnel, coal mine, deformation detection

## Abstract

In recent years, the deformation detection technology for underground tunnels has played a crucial role in coal mine safety management. Currently, traditional methods such as the cross method and those employing the roof abscission layer monitoring instrument are primarily used for tunnel deformation detection in coal mines. With the advancement of photogrammetric methods, three-dimensional laser scanners have gradually become the primary method for deformation detection of coal mine tunnels. However, due to the high-risk confined spaces and distant distribution of coal mine tunnels, stationary three-dimensional laser scanning technology requires a significant amount of labor and time, posing certain operational risks. Currently, mobile laser scanning has become a popular method for coal mine tunnel deformation detection. This paper proposes a method for detecting point cloud deformation of underground coal mine tunnels based on a handheld three-dimensional laser scanner. This method utilizes SLAM laser radar to obtain complete point cloud information of the entire tunnel, while projecting the three-dimensional point cloud onto different planes to obtain the coordinates of the tunnel centerline. By using the calculated tunnel centerline, the three-dimensional point cloud data collected at different times are matched to the same coordinate system, and then the tunnel deformation parameters are analyzed separately from the global and cross-sectional perspectives. Through on-site collection of tunnel data, this paper verifies the feasibility of the algorithm and compares it with other centerline fitting and point cloud registration algorithms, demonstrating higher accuracy and meeting practical needs.

## 1. Introduction

Coal mine tunnel deformation detection is a critical link to ensuring the safety of coal mine workers. In order to ensure the stability of the tunnel structure, indicators such as cross-section [1], convergence [2], and centerline settlement [2] are key factors in measuring tunnel deformation. Tunnel deformation monitoring involves analyzing the health status, change patterns, and development trends of the tunnel. The dim and humid conditions inside coal mine tunnels impose significant limitations on long-term underground operations. Coal mine tunnels typically exhibit a narrow and tubular structure, with tunnel cross-sections including rectangular, arched, and trapezoidal shapes. As anchor rods and surrounding rock control technologies such as support systems mature in underground coal mines, the predominance of arched tunnels in the early stages has gradually shifted towards rectangular ones. This study focuses on rectangular tunnels.

Deformation analysis of tunnels is an important indicator for evaluating tunnel safety. Currently, there are three main methods for deformation analysis: The first method involves using direct measurement or high-precision measurement methods to obtain the true values of points or surfaces within the tunnel. This allows for a comparison between the measurement data and the actual data of the tunnel to analyze the local deformation of the tunnel. However, this method is limited by the actual measurement points, reflecting only limited local deformations. Additionally, it requires a significant amount of manpower and time, making timely and accurate deformation detection difficult. The second method involves comparing measurement data with tunnel construction design values to obtain the overall and local deformations of the tunnel. Due to the complex geological conditions in underground coal mines and variations in the surrounding rock environment during construction, discrepancies between the actual tunnel and the design drawings lead to inaccurate deformation detection. The third method involves repeatedly collecting three-dimensional structural data from different periods within the tunnel. By comparing data from different periods, deformations at all positions of the tunnel can be accurately reflected. Currently, the main measurement method for this approach is laser scanning, which identifies locations with significant tunnel deformations by comparing point cloud data from different periods. Based on three-dimensional laser scanning technology, deformation detection algorithms are divided into direct comparison methods of point cloud data and indirect comparison methods using point cloud fitting models. Since fitting models cannot avoid fitting errors, this paper aims to improve the accuracy of the algorithm by using point cloud data from different periods to calculate tunnel deformations.

Currently, tunnel deformation analysis methods based on three-dimensional laser scanning are mainly divided into stationary and mobile LiDAR systems. Stationary LiDAR systems offer high measurement accuracy but require long operation times and significant manpower costs. Therefore, considering underground coal mine safety, mobile LiDAR systems are gradually becoming a popular measurement method. Mobile three-dimensional laser scanning has higher measurement efficiency and can reflect specific deformations and deformation trends of the tunnel. However, it imposes high demands on the registration of point cloud data from different periods.

To address these challenges, this paper proposes a method based on mobile laser scanning technology. It utilizes the centerline registration method to register the point cloud data of tunnels from different periods into a unified coordinate system. Then, it employs the k-nearest neighbor method to find corresponding point pairs in point clouds from different periods, analyzing the deformation of both the overall and local cross-sections of rectangular tunnels. The effectiveness of this method is verified using actual tunnel point cloud data, demonstrating its compliance with point cloud measurement accuracy and meeting practical production needs. The innovation of this paper lies in the following:This paper presents an algorithm for extracting the centerline of rectangular tunnel point clouds. By projecting 3D point clouds onto two-dimensional planes in different directions, the contours of rectangular tunnel point clouds are calculated based on the statistical distribution pattern of points on the plane. The centerline of the entire tunnel is determined based on the characteristics of the center points in the rectangular cross-section;A method for point cloud registration based on the centerline of the tunnel is proposed in this paper. Similarly, employing a projection method, the centerline of the tunnel is projected onto different planes to calculate the rotation and translation matrices in the x, y, and z directions for the centerline of the tunnel at different time periods. Consequently, the entire tunnel point cloud is registered;An analysis method for tunnel deformation based on both local and global features is proposed in this paper. Utilizing the K-nearest neighbor (KNN) algorithm to find neighboring points, corresponding point pairs in point clouds at different time periods are determined. By computing the difference in distances between point pairs, the deformation quantity of each point is determined, thereby obtaining the analysis of the overall and cross-sectional deformation of the tunnel.

## 2. Related Work

### 2.1. Laser Point Cloud Alignment Method

Three-dimensional LiDAR point cloud registration is crucial for LiDAR odometry and LiDAR SLAM. It is the fundamental basis for simultaneously localizing and mapping with LiDAR sensors. Achieving data association is the most commonly used method in LiDAR-SLAM. Existing 3D LiDAR point cloud registration methods are mainly divided into three categories: point-based methods, distribution-based methods, and feature-based methods.

The Iterative Closest Point (ICP) algorithm, based on point-based methods, is currently one of the most researched, widely applied, and mature algorithms. In the ICP algorithm, transformation between adjacent point clouds is computed iteratively by minimizing a distance function. Besl [3] demonstrated that the ICP algorithm consistently converges monotonically to local minima under the strong assumption that the number and relationships of corresponding point pairs remain unchanged during the iterative process. To enhance the accuracy, efficiency, and robustness of point cloud registration algorithms, researchers have proposed various improved ICP algorithms. Censi [4] introduced a “point-to-point” ICP algorithm that starts searching for correspondences from geometric points closest to the three-dimensional shape. Low [5] introduced a “point-to-plane” ICP algorithm, which estimates the target model as a plane to obtain distance data. Chen [6] proposed the P2Pl-ICP algorithm, which uses the point-to-plane distance instead of the point-to-point distance as the error metric, thus improving the algorithm’s robustness. Segal [7] introduced the Generalized ICP (GICP) algorithm, which leverages the local continuity of point cloud surfaces, approximating the surface shape around point clouds as a planar patch and considering the sensor’s noise model to effectively reduce the impact of mismatches. Although this method exhibits strong effectiveness and robustness among many improved ICP algorithms, its performance in scenarios of coal mine tunnels is not as good as the original ICP algorithm.

The Normal Distributions Transform (NDT) algorithm, based on distribution-based methods, is a typical approach initially proposed for two-dimensional LiDAR point cloud registration. Laser point clouds are represented by a set of Gaussian distributions with different probability density functions. To avoid incorrect correlation issues between data, this method provides a segmented smoothed normal distribution representation of laser scan data. Its major advantage lies in not requiring explicit correspondences between features or points. Therefore, the algorithm demonstrates good robustness. Magnusson [8] proposed a P2D-NDT scan matching method, extending NDT from 2D to 3D. This algorithm divides the reference frame LiDAR point cloud into small three-dimensional grid cubes, calculates the probability density function for each grid based on the shape containing internal points, and solves the relative pose transformation problem by maximizing the points scanned from the current frame LiDAR onto the reference frame surface. Magnusson [9] further demonstrated through practical mining trials that NDT exhibits stronger adaptability, accuracy, and robustness compared to ICP. However, this method relies on NDT scan registration for localization and mapping, leading to inevitable error accumulation in LiDAR odometry with increasing registration processes.

Feature-based methods extract simple features from LiDAR point clouds for efficient feature matching, improving point cloud registration efficiency. These extracted features are then used to determine the relative pose changes between points in point cloud registration. These simple features can be selected as points, lines, planes, or their combinations. Point feature-based point cloud registration methods, which find corresponding points by extracting feature points, are most suitable for two-dimensional point cloud registration. However, many feature descriptors are designed for specific environmental conditions. Line feature-based point cloud registration methods exhibit many simple and efficient line features in indoor scenes. Liu [10] proposed a widely used segment merging method for line feature-based point cloud registration in two-dimensional LiDAR point cloud data. Point cloud registration methods based on face features complement the shortcomings of extracting point and line features by detecting a large number of planes or surface features within a region, reasonably utilizing the extracted features for data association and point cloud matching. For scenes containing curved surfaces, Nobili [11] used voxel filters to uniformly downsample raw 3D LiDAR point cloud data, registered point clouds on feature planes, and eliminated irrelevant interference outside the feature planes. It is noteworthy that Zhang [12] proposed a typical approach to obtain accurate results and reduce computational complexity. This method matches feature points with edges or surfaces by extracting edge and surface features from the environment, thereby achieving inter-frame point cloud registration. However, while feature-based methods demonstrate good performance in autonomous robot localization and mapping, they may produce errors due to the lack of geometric features or feature degradation in certain scenes, significantly affecting point cloud registration accuracy.

### 2.2. Method for Tunnel Deformation Detection Based on Laser Scanning

Tunnels are a type of special underground engineering characterized by their elongated shape, complex construction environment, and the need to consider various factors. The common three-dimensional laser scanning methods are classified into stationary and mobile types. Stationary 3D scanners have been proven effective for tunnel deformation detection [13,14,15]. Han [16] proposed an automatic and efficient method for estimating tunnel centerlines and cross-sectional planes. This method estimates the tunnel centerline by projecting the three-dimensional point cloud onto a horizontal plane to generate a binary image. Based on the tunnel boundary points and centerline, the cross-sectional planes are estimated and further adjusted by projecting nearby points onto the adjusted plane to generate the final cross-section. Although the authors improved the efficiency of measurement and data processing, they were unable to extract continuous contours due to the lack of a parameter equation for the tunnel centerline. Additionally, this method is sensitive to non-lining points, namely, pipes and equipment attached to the lining. To overcome the limitations of this method, which include fitting straight lines, transition curves, and curves to different segments of the tunnel centerline, Kang [1] fitted them jointly and then applied global least squares adjustment to maintain consistency between adjacent fitting models. To handle missing data, Kang used surface interpolation algorithms to recover blank areas in the tunnel profile. For accurate orientation of cross-sectional planes, Cheng [17] performed two optimizations on the initial sections using the total least squares method and the Rodrigues rotation formula. Li [18] utilized stationary laser scanners in coal mine tunnels, deploying target balls in the tunnels to register point clouds from different periods in the same coordinate system, enabling continuous deformation analysis of section point clouds at the same location over time, thus obtaining a more comprehensive and intuitive understanding of overall tunnel deformation. However, the use of stationary 3D scanning in coal mine tunnel scenes still has certain limitations, requiring high demands on the tunnel environment and measurement time.

Currently, the detection of tunnel deformation based on mobile 3D scanning technology has become a hot area. Lu [19] compared the accuracy and efficiency of stationary and mobile laser scanning technologies. Stationary 3D scanning has higher accuracy, with errors controlled at the millimeter level, while mobile 3D scanning has errors controlled at the centimeter level. However, stationary 3D scanning operates at a slower speed. To ensure scanning accuracy, the measurement range for each station of stationary scanning is usually between 15 and 25 m. Without considering rest time for personnel and equipment, the average measurement along the tunnel is 2.8 m per minute. Mobile 3D scanning technology has higher operational efficiency, measuring approximately 33 m forward per minute on average. Typically, a tunnel has a long distance. For example, for a tunnel that is 3 km long, using stationary 3D scanning technology for tunnel monitoring would require continuous operation for approximately 18 h to complete one full measurement of the tunnel, whereas mobile 3D scanning would require approximately 1.5 h. Therefore, mobile 3D scanning is more suitable for practical tunnel deformation monitoring in mining production. Cheng [20] conducted deformation monitoring and analysis on a section of a subway operating in Hangzhou, China, using a Swiss company’s GRP5000 laser scanning holographic system. They fitted the cross-sectional profile using the RANSAC-LSM algorithm and analyzed the ellipticity and convergence deformation of the tunnel. The algorithm validated that the results of mobile scanning were basically consistent with those of total station measurements. Zhou [21] used the Riegl VZ-1000 mobile laser scanner to collect point cloud data of tunnel surrounding rock. By comparing with conventional deformation measurement methods, they found that the deformation of tunnel surrounding rock is closely related to its stress condition. They also validated the measurement accuracy of mobile 3D laser scanning technology, meeting the requirements of tunnel deformation monitoring in underground coal mines. Ye [22] proposed a tunnel point cloud deformation detection method based on mobile laser scanning technology. They used point cloud data collected by a vehicle-mounted laser radar to fit each cross-section of the tunnel into an ellipse and compared it with the standard circle of the tunnel design section to analyze the external forces of tension and compression acting on the tunnel.

Regardless of whether based on stationary or mobile three-dimensional laser scanning technology, current deformation detection of tunnels largely relies on cross-sectional analysis of the tunnels. The analysis method based on tunnel cross-sections always entails some risk zones between adjacent sections that cannot be accurately tracked, where deformations may easily be overlooked. Fekete [23] conducted triangulated mesh modeling of tunnel surface point clouds, transforming discrete points into continuous surfaces. Dimitrov [24] utilized Non-Uniform Rational B-Spline (NURBS) functions to fit arbitrary three-dimensional geometries within point clouds. Nahangi [25] proposed a method for skeletonizing point cloud three-dimensional models, automatically quantifying the three-dimensional structure of point clouds. Van [26] approximated the scanning surface of a bore tunnel using a cylindrical model and conducted point-wise deformation analysis by comparing surface patches. Although methods based on tunnel models can detect minor differences by comparing tunnel models generated at different times, these methods are highly sensitive to surface point cloud noise, leading to increased errors. Moreover, fitting with models such as planes, cylinders, or spheres reduces algorithmic computational efficiency, failing to meet the deformation detection requirements of large-scale point clouds.

### 2.3. Analysis Method of Deformation in Coal Mine Underground Tunnels

In underground coal mine tunnels, there is a significant accumulation of coal piles, rock masses, etc. The surrounding rocks are prone to deformation, and the cross-sectional shapes include rectangles, arches, trapezoids, and other shapes, making the environment more complex compared to urban underground tunnel projects. The most common detection methods include the crisscross measurement [27] and the roof separation meter method [28], which are based on the principles of traditional mechanical transmission. These methods transmit the deformation of the tunnel surface to the deformation of ropes or metal components through devices, which are then displayed using measuring instruments such as rulers.The mechanical three-coordinate measuring machine, by placing the probe on the surface of the object to be measured, obtains three-dimensional data at the contact point of the object. Measuring machines such as the Zeiss gantry measuring machine, MICRO-HITE measuring machine, RPS articulated arm three-dimensional coordinate measuring machine, and China DeRen articulated arm three-dimensional coordinate measuring machine can all achieve a measurement progress of 1 μm. However, traditional inspection methods can cause damage to the structure of tunnels, with the characteristic of the inspection points being distributed discretely, unable to ensure continuous coverage of the overall surface of the tunnel. Analysis of tunnel cross-sections and overall trends requires the use of interpolation fitting, which can easily lead to significant errors. Therefore, contact measurement technology is not suitable for the analysis of large-scale overall deformation of coal mine tunnels.

With the emergence of indirect measurement technology, level instruments [29,30] and total stations [31,32] are widely used in tunnel deformation detection. Ye [33] designed a set of equipment that can replace the traditional cross measurement method using a laser rangefinder, which monitors the surface displacement of the fixed positions of the tunnel roof, floor, and side walls with a single laser rangefinder. Song [34] used geological radar to study the laws of tunnel deformation and geological condition changes.

These methods suffer from long measurement times and high work intensity and can damage the health of the operator. Therefore, three-dimensional laser scanning technology for tunnel deformation detection has become popular research because of its high efficiency. Dai [35] utilized the alphashape algorithm to fit rectangular cross-sections of tunnels and analyzed tunnel deformation parameters by calculating the difference between points and the fitted lines. While this algorithm can intuitively represent tunnel deformation parameters, the accuracy of deformation parameters and point cloud fitting are directly related. Liu [36] proposed a method for detecting and monitoring the deformation of coal mine underground tunnel roofs through the establishment of a matching section model, analysis of the deformation of the point cloud distance model for each section, and control of the measurement error within 3 mm. They also analyzed the mechanical structure of surrounding rocks and the influence of lithology on tunnel changes. However, these algorithms require the use of a stationary LiDAR system, which is not suitable for underground mining faces with complex environments.

Yu [37] utilized the principle of linear scanning, combined with calibrated cameras, to design and develop a vehicle-mounted deformation measurement system for coal mine tunnels. In simulated tunnel environments, the system verified the accuracy of deformation detection algorithms by setting its own deformation true values, with errors controlled within 7 mm. While this algorithm adapts well to environments with good lighting conditions such as coal mine main tunnels and transport tunnels, it struggles to adapt in mining faces due to insufficient lighting and high levels of airborne dust. Rong [38] researched methods for extracting boundaries of underground coal mine tunnels, designing a vehicle-mounted 3D laser scanning system. They employed a method of target ball placement for point cloud stitching and coordinate transformation, enabling three-dimensional modeling and deformation analysis of the entire tunnel. Deformation errors at the intersection of the coal walls and roof were controlled within 8 cm, providing a reference for the cutting of coal mining machines. Qi [39] utilized a method of establishing three-dimensional maps using vehicle-mounted laser point clouds to monitor deformation of underground comprehensive mining faces in coal mines. Experimental data showed that this algorithm ensured that measurement errors were controlled within 19 cm. Zou [40] addressed large-scale coal mine tunnel point clouds, using integrated factor graph optimization theory and mobile robots to construct three-dimensional tunnel maps, detecting trends in tunnel changes. They affirmed the INE-SLAM algorithm, with an average error control of 6.5 cm. Du [41] focused on the deformation of coal mine slopes, utilizing point cloud data for three-dimensional modeling and polygonal partitioning of the model. By comparing the centroid offsets of polygons, they analyzed the displacement deformation of coal mine slopes, consistent with real conditions. Wang [42] proposed transparency and intelligence-based mining with transparency as the core concept, focusing on deformation safety of comprehensive mining faces. Gao [43] proposed a prevention and control technology for the deformation of soft rock in tunnel roofs. These algorithms typically utilize existing three-dimensional spatial data to establish three-dimensional models and analyze deformations by comparing actual measured point cloud data with the model. In underground coal mine environments, where actual deformation data are often lacking, fitting three-dimensional models using point cloud data introduces new errors. Therefore, direct comparison of point cloud data from different periods to identify locations with significant deformation enables the reflection of overall tunnel deformation trends and local deformations, thereby drawing the attention of management personnel.

## 3. Methods

This article presents a coal mine tunnel point cloud deformation detection algorithm based on mobile three-dimensional laser scanning technology. The algorithm is mainly divided into three steps: tunnel centerline extraction, point cloud registration, and deformation detection, as shown in Figure 1. Two sets of point cloud data collected at different times are projected onto different two-dimensional planes. The boundary feature lines of the tunnels are extracted by utilizing the distribution probability of the point cloud in the two-dimensional plane, thereby calculating the centerline of the tunnel. The rotation and translation matrix for registering the centerlines of the two tunnels is computed, thereby registering the two sets of point cloud data to the same coordinate system. Finally, this article analyzes the deformation of the coal mine tunnel from overall and cross-section perspectives.

### 3.1. Extraction of the Tunnel Centerline

The centerline of a tunnel refers to the central axis of an underground tunnel in a coal mine. In the engineering design and construction of tunnels, it is usually necessary to determine the centerline of a tunnel based on the actual geological conditions and parameters such as the slope, width, and height of the tunnel. At the same time, the centerline of the tunnel is continuously used to control the excavation and construction process of the tunnel. Therefore, accurate tunnel centerlines are crucial for coal mine tunnels. The underground tunnel environment in coal mines is complex, and equipment such as wires, air ducts, and belt conveyors can affect the quality of the underground tunnel point cloud data. Moreover, the tunnel cross-section is not strictly rectangular, so the centerline cannot be simply extracted by the center point of the minimum bounding rectangle.

This article proposes a tunnel centerline extraction method based on density analysis to address this situation. This method uses slicing and projection to reduce the three-dimensional point cloud to a two-dimensional scatter plot. By statistically analyzing the distribution quantity and density of points in the horizontal and vertical directions of the two-dimensional image and using kernel density analysis and line extraction methods, the side lines and top and bottom lines of the tunnel are extracted separately. Finally, the boundary lines of the entire tunnel are fitted using a non-uniform rational B-spline function to calculate the centerline of the tunnel.

#### 3.1.1. Point Cloud Dimensionality Reduction

Most tunnels are distributed along straight lines. Firstly, the PCA method is employed to extract the main direction of the tunnel. The extension direction of the tunnel is rotated to overlap with the x-axis direction, facilitating subsequent point cloud data processing. At the same time, cross-sectional point clouds perpendicular to the extension direction of the tunnel are extracted as tunnel slices. To extract richer point cloud distribution features, the sliced point clouds are projected onto the xoy plane and the yoz plane, which represent the cross-sectional view and the plan view of the slices, as is shown in Figure 2. The colored part represents a slice of the rectangular tunnel point cloud, which is perpendicular to the extension direction of the tunnel. The blue plane in the figure represents the scatter plot mapped to the yoz plane, and the red plane represents the scatter plot mapped to the xoy plane. By analyzing the distribution density of two-dimensional scatter points in the directions shown in Figure 2a,b, the boundary points of the tunnel are extracted.

#### 3.1.2. Extraction of the Tunnel Roof and Floor Lines

Due to the consistent distribution of point clouds on the top and bottom plates of the tunnel, the sliced cross-sectional point cloud diagram better clusters the point clouds of the top and bottom plates together along the direction of the tunnel extension, thereby better reflecting the higher distribution density of the point clouds of the top and bottom plates compared to equipment points and surrounding noise points in space.

By statistically counting the number of points in the longitudinal direction of the sliced cross-sectional point cloud and analyzing the degree of point aggregation, the value most concentrated in the z-axis coordinate distribution of the sliced point cloud is obtained. Since there is no prior experiential knowledge of the probability density of point cloud distribution, neither parameter estimation nor semi-parameter estimation methods are applicable to such problems. Therefore, this paper employs the kernel density estimation method, leveraging the advantages of non-parametric estimation to query the kernel density peaks at larger and smaller values along the longitudinal axis, which are the points where the point cloud distribution is most concentrated near the top and bottom plates, thereby determining the z-axis coordinates of the top and bottom plates at the sliced section of the tunnel.

The alley is vertically divided into *n* equal parts for slicing. The point cloud count of each part is statistically analyzed using histograms, and the aggregation level of the point cloud is analyzed using one-dimensional kernel density estimation. The point cloud count of each part from top to bottom is denoted as $X=x_{1},x_{2},\ldots ,x_{n}$, with each *x* being independently distributed. Therefore, the density function of X follows the distribution as shown below: (1)$$f\left(x\right)=\frac{1}{n}\sum_{i=1}^{n}K_{h}\left(x-x_{i}\right)=\frac{1}{nh}\sum_{i=1}^{n}K\left(\frac{x-x_{i}}{h}\right)$$
K(x) is the kernel function, where *h* is the width of the kernel function window.

The Gaussian kernel function is one of the commonly used kernel functions in kernel density estimation, and it is a typical smoothing function. The expression of the Gaussian kernel function is shown below: (2)$$K\left(x\right)=\frac{1}{\sqrt{2\pi }}e^{-\frac{x^{2}}{2}}$$
where *x* is the variable. The basic idea of Gaussian kernel density estimation is to obtain the probability density estimation value of each data point by weighted averaging of the data points within a certain range around it. Specifically, for each data point *x*, a estimation range with a width of *h* is constructed around it. Then, for each data point *x*, the weights of all data points around it are calculated, and the weights are determined by the Gaussian kernel function.

The Gaussian kernel function has the characteristics of smoothness and symmetry. It filters out noise around the point cloud of tunnel boundary points well and accurately estimates the coordinates of points where the point cloud density is high at tunnel boundaries, without making distribution probability assumptions about the original point cloud slice data in advance, demonstrating high applicability and flexibility. It helps to obtain a smooth probability density function in density estimation, can overcome the influence of point cloud edge noise, and find areas of high point cloud density. Moreover, the Gaussian kernel function is symmetric, K(u)=K(−u), which is suitable for the disorderliness of point clouds. The shape of the Gaussian kernel function is controlled by the bandwidth parameter. The larger the bandwidth is, the wider the range of the kernel’s effect and the smoother the estimation.

As shown in Figure 3, according to the distribution of the number of section point clouds in the horizontal direction, the two z-values with the densest distribution of point clouds are selected in this paper as the top and bottom plate lines of the roadway section. The left subplot is the cross-sectional scatter plot of the sliced point cloud of the tunnel, where the red line represents the detected tunnel floor line, and the green line represents the detected tunnel roof line. The right subplot shows the density estimation of the z-coordinate of the sliced point cloud of the tunnel obtained through one-dimensional kernel density detection. The red point represents the first density peak point representing the floor, and the green point represents the last density peak point representing the roof.

#### 3.1.3. Extraction of the Tunnel Wall Line

Due to the presence of numerous wires, anchor rods, and other equipment distributed on both sides of the tunnel in the cross-section, which affects the extraction of tunnel boundaries, methods such as two-dimensional linear kernel density analysis, Canny edge detection, and Hough transform are employed on the overhead view of the slice point cloud to identify the left and right boundaries of the tunnel.

This article first divides the two-dimensional slice plane space into n×n blocks, counts the number of point clouds in each block, and calculates the kernel density in both horizontal and vertical directions for each block, resulting in a kernel density map. When performing two-dimensional kernel density estimation, this study employs linear kernel density estimation. The expression of the linear kernel function is as follows:(3)$$K\left(x\right)=\left\{\begin{array}{c} 1-|x|,\;\;|x|<1\\ 0,\;\;others \end{array}\right.$$

The linear kernel function can effectively smooth the data, distinguishing different density boundaries more prominently than the Gaussian kernel function, resulting in a density distribution plot with clearer linear patterns, which prepares for subsequent line detection.

The new density distribution plot undergoes Canny edge detection and Hough transform. Canny edge detection and Hough transform are common methods for feature extraction in image processing and computer vision. Canny edge detection typically identifies image boundaries by locating positions with the maximum pixel value gradients in the image. The steps for Canny edge detection used in this study involve first converting the kernel density plot to a grayscale image, then computing the gradients along the *x* and *y* axes of the image and calculating the combined gradient direction. The gradient calculation formula is $G=\sqrt{G_{x}^{2}+G_{y}^{2}},\theta =\arctan\frac{G_{y}}{G_{x}}$, where $G_{x}$ is the magnitude of the horizontal gradient, $G_{y}$ is the magnitude of the vertical gradient, and θ is the gradient direction. Subsequently, non-maximum suppression is used to eliminate some false boundary points, and finally, by determining the maximum and minimum values of the double threshold, the edges present in the kernel density plot are identified, with edge position pixels marked as white and non-edge position pixels marked as black.

Next, the new image after Canny edge detection undergoes the Hough transform. The Hough transform was originally proposed by Paul Hough in 1962 to detect lines in images. The steps for using the Hough transform in this study involve first treating the pixels of the image as squares in parameter space and then determining the squares passed through by each white pixel in the parameter space through a mapping relationship. Secondly, the occurrences of each square in the parameter space are counted, selecting squares with occurrences greater than a certain threshold as representative squares for lines. Finally, the parameters of squares representing lines in parameter space are taken as the parameters for lines in the image.

As shown in Figure 4, the left subplot depicts a top-down view of the lane point cloud slice, where the red line segments represent the extracted left boundary line and the green line segments represent the extracted right boundary line. The middle subplot displays a two-dimensional linear kernel density estimation of the sliced point cloud top-down view, where the red dots depict the effect of magnifying the sliced point cloud along the x-axis and the blue portion represents the values estimated by the kernel density. The right subplot illustrates the effect of Canny edge detection and Hough line extraction on the kernel density plot in the middle subplot, where the red represents the first line detected from top to bottom, representing the left boundary line, and the green represents the last line detected, representing the right boundary line.

#### 3.1.4. Curve Fitting

After extracting the top and bottom plates and two sides of the tunnel slice, respectively, calculate the midpoint of the top and bottom plates and the midpoints of the two sides to obtain the fitted slice point cloud center. Tunnel slicing is a process of discretizing the entire tunnel. After obtaining each discrete center point, it is necessary to perform curve fitting to obtain a continuous tunnel centerline. The curve fitting method selected in this paper is NURBS curve fitting.

NURBS (Non-Uniform Rational B-Splines) curves are a commonly used method of curve representation in computer graphics. Its main feature is that control points can be non-uniformly distributed. Due to the construction of chambers on both sides of the tunnel to ensure operational safety, the tunnel centerline is not completely straight. In the method based on slices, at the chamber of the tunnel, the spacing between control points is large and sparse. NURBS curves can make good use of non-uniform control points to achieve better curve fitting effects. NURBS curves use rational B-splines as a foundation. The introduced weight parameter brings greater flexibility to the curve, allowing it to represent various curve shapes. NURBS curves use B-spline functions as a foundation, giving them strong local control properties, meaning that moving or adding control points only affects the local region of the curve. The algorithm combines the advantages of Bézier curves and spline curves and has strong expressive power. The steps of NURBS curve calculation are as follows: Each slice’s center is determined as the control point P of the NURBS curve, and the corresponding weight w is given for each control point. The weight of the control point represents the degree of adhesion of each control point. Identify the nodes u of the basis functions, dividing the parameter space into m nodes, ensuring that the values within the nodes are monotonically increasing.
(4)$$P=\left[P_{0},P_{1},P_{2},\ldots ,P_{n}\right],W=\left[W_{0},W_{1},W_{2},\ldots ,W_{n}\right],u=\left[u_{0},u_{1},u_{2},\ldots ,u_{m}\right]$$

Calculate the B-spline function based on the specified domain and curve order. Based on the determined parameters above, calculate the formula of the NURBS curve. Construct the final NURBS curve by computing the values of parameters in each defined domain, as shown in the Figure 5. The blue points represent the control points of the curve, the red points denote the knots, and the black curve represents the NURBS fitted curve.
(5)$$N_{i,0}\left(u\right)=\left\{\begin{array}{rr} 1, & u_{i}\leq u\leq u_{i+1}\\ 0, & others \end{array}\right.$$
(6)$$N_{i,k}\left(u\right)=\frac{u-u_{i}}{u_{i+k-1}-u_{i}}N_{i,k-1}\left(u\right)+\frac{u_{i+k+1}-u}{u_{i+k+1}-u_{i+1}}N_{i+1,k-1}\left(u\right)$$

### 3.2. Tunnel Centerline Alignment Algorithm

In terms of deformation monitoring, point cloud registration is crucial. Therefore, this paper proposes a method to align the point cloud of the roadway based on the centerline extracted from the scan. The main idea of this method is to use the three views of the roadway to realize the alignment of the overall point cloud of the roadway by projecting the centerline of the roadway to different planes and aligning them.

#### 3.2.1. Point Cloud Alignment Based on Principal Component Analysis (PCA)

Using principal component analysis for point cloud coarse registration involves deriving three characteristic vectors of the tunnel point cloud through principal component analysis, which serve as the new coordinate system for the point cloud, i.e., performing a base transformation on the original point cloud. The direction of maximum variance represents the extension direction of the tunnel point cloud, namely, the x-axis direction; the direction of the second largest variance represents the y-axis direction; and the direction of the smallest variance represents the z-axis direction. Principal component analysis is separately applied to the original point cloud data and the point cloud data to be registered to obtain three characteristic vectors. Rotation and translation matrices are calculated to align the characteristic vectors of the point cloud to be registered with those of the original point cloud. The calculated rotation and translation matrices are applied to the point cloud to be registered to perform rigid transformation, achieving coarse registration of the two sets of point cloud data.

The advantages of the principal component analysis method are as follows: (1) a certain resistance to noise, unaffected by the quality of the point cloud data; (2) the orthogonality of the reduced characteristic vectors, which can effectively restore the spatial relationship of the point cloud; and (3) the high computational efficiency of the algorithm, making it suitable for analyzing three-dimensional point cloud data.

#### 3.2.2. Alignment of Tunnel Cross-Sections

The cross-section of a tunnel refers to the profile perpendicular to the centerline of the tunnel. Cross-sections primarily display the geometric characteristics such as the shape and dimensions of the tunnel, which are crucial for designing and planning tunnel projects as well as assessing the stability and safety of underground spaces. This paper preprocesses the point cloud data of the tunnel, aligning the centerline of the tunnel with the X-axis. Therefore, the registration of tunnel cross-sections mainly involves rotation and translation in the YOZ plane.

Utilizing the segmentation algorithm mentioned in Chapter Three of this paper, the point cloud of the tunnel floor is segmented. Using the RANSAC plane fitting algorithm, rotate the normal vector of the base plane of the point cloud to be registered to the normal vector of the base plane of the target point cloud. The rotation center is at the center of the point cloud’s minimum bounding box, as shown in the Figure 6. The RANSAC plane fitting algorithm is one of the most commonly used algorithms in point cloud processing. Its main principle is to randomly select a subset from the samples and use the least squares estimation algorithm to calculate the model parameters. Compute the deviation of all samples from this model and compare it with a pre-set threshold. When the deviation is less than the threshold, the sample point is marked as an inlier; otherwise, it is marked as an outlier. Record the current number of inlier points and repeat the above process. In each iteration, record the current best model parameters that show the most inlier points. Calculate the iteration termination criterion based on the desired error rate, the optimal number of inliers, the total number of samples, and the current iteration count. Decide whether to terminate the iteration based on the termination criterion. If the iteration terminates, the best model parameters are then considered as the final estimated model parameters.

#### 3.2.3. Alignment of the Longitudinal Section of a Tunnel

The longitudinal section of a tunnel refers to the profile along the main direction of the tunnel. The longitudinal section mainly shows the extension, distribution, and overall shape of the tunnel, which play a very important role in the overall engineering design, planning, and analysis of the tunnel. The registration of the longitudinal section of the tunnel mainly refers to rotation and translation in the xoz plane. Since the top and bottom plates of the underground tunnel can be approximately seen as a plane without significant undulations, the centerline of the tunnel extracted using the method mentioned earlier can be approximated as a straight line in the xoz plane. In this paper, the least squares method is used to fit a straight line to the x and z values of the centerline, calculate the rotation matrix needed to rotate the fitted line of the point cloud to the fitted line of the target point cloud, apply this rotation matrix to the point cloud, and obtain the longitudinally registered tunnel point cloud data. The method is shown in Figure 7, where the red line is the yellow point cloud centerline and the green line is the blue point cloud centerline. Using the intersection of the two fitted centerlines as the center of rotation, the rotation matrix required to rotate the two centerlines to overlap is calculated to align the two tunnel point clouds.

The main principle of least squares line fitting is to minimize the sum of squared errors for discontinuous discrete points to determine the parameters of the line to be fitted. In this paper, the x values of the tunnel centerline are taken as the independent variable, and the z values of the tunnel centerline are taken as the dependent variable, to obtain the coordinates of several two-dimensional plane points, $\left(x_{1},z_{1}\right),\left(x_{2},z_{2}\right),\ldots ,\left(x_{n},z_{n}\right)$. Assuming the fitted straight line is Z=ax+b, for each $x_{i}$, the error function of the estimates and observations is as follows: (7)$$f\left(x\right)=\sum_{i=0}^{n-1}{\left(Z_{i}-z_{i}\right)}^{2}=\sum_{i=0}^{n-1}{\left(ax_{i}+b-z_{i}\right)}^{2}$$

Find the values of a and b when f(x)=min. Find the partial derivatives of the error function f(x)=min with respect to a and b, respectively, and calculate the values of a and b when the partial derivatives are equal to 0. The formula is as follows: (8)$$\frac{\partial f}{\partial a}=2\left(a\sum_{i=0}^{n-1}{x_{i}}^{2}+b\sum_{i=0}^{n-1}x_{i}-\sum_{i=0}^{n-1}x_{i}z_{i}\right)=0$$
(9)$$\frac{\partial f}{\partial b}=2\left(a\sum_{i=0}^{n-1}x_{i}+bn-\sum_{i=0}^{n-1}z_{i}\right)=0$$
(10)$$a=\frac{n\sum_{i=0}^{n-1}x_{i}z_{i}-\sum_{i=0}^{n-1}x_{i}\sum_{i=0}^{n-1}z_{i}}{n\sum_{i=0}^{n-1}{x_{i}}^{2}-{\left(\sum_{i=0}^{n-1}x_{i}\right)}^{2}},\;b=\frac{\sum_{i=0}^{n-1}z_{i}\sum_{i=0}^{n-1}{x_{i}}^{2}-\sum_{i=0}^{n-1}x_{i}\sum_{i=0}^{n-1}x_{i}z_{i}}{n\sum_{i=0}^{n-1}{x_{i}}^{2}-{\left(\sum_{i=0}^{n-1}x_{i}\right)}^{2}}$$

Compared to the RANSAC line fitting algorithm, the least squares method calculates the line to be fitted using all selected points, while the RANSAC algorithm removes some outliers and fits the line through multiple iterations for inliers with distances to the line smaller than a threshold. As a result, the line fitted by RANSAC may have some error compared to the line fitted by the least squares method.

#### 3.2.4. Alignment of Tunnel Planes

The tunnel plan is a map that shows the layout and features of the tunnel in the horizontal direction. This type of diagram displays the planar shape, dimensions, connectivity, and other detailed information of the tunnel. In the actual mining operation process, the tunnel plan is the most commonly used engineering drawing to guide operations. Since the development and mining tunnels are usually equipped with galleries to ensure the safety of coal miners during the mining process, the projection on the xoy plane cannot simply be fitted into a straight line, but rather fitted into a curve using the method described in the preceding section. This paper utilizes the idea of the nearest point iteration algorithm, combined with the point cloud correlation coefficient, to calculate the rotation and translation matrices between the point clouds $P:p_{1},p_{2},\ldots ,p_{n}$ and the target point cloud $Q:q_{1},q_{2},\ldots ,q_{n}$.

The nearest point iteration algorithm [3] is an algorithm used to minimize the difference between two point clouds. In each iteration, the algorithm selects the closest point as the corresponding point and calculates the transformation (R,T) to minimize the equation: (11)$$E\left(R,T\right)=\sum_{i=1}^{N_{p}}\sum_{j=1}^{N_{q}}w_{i,j}{∥p_{i}-\left(Rq_{j}+T\right)∥}^{2}$$
where $N_{r}$ and $N_{t}$ are the number of points in the point clouds P and Q to be registered, respectively, and $w_{ij}$ is the weight of point matching. To improve efficiency, there have been many optimizations to the ICP optimization algorithm. However, the ICP algorithm still suffers from the problem of falling into local optimal solutions, affecting the registration accuracy of the algorithm [44,45]. Based on this algorithm, this paper proposes a new two-dimensional point cloud registration algorithm that aligns the two curve point clouds by separately calculating the rotation and translation matrices. The method is shown in Figure 8; the green centerline and the red centerline are made to overlap as much as possible by panning and rotating operations.

When calculating the rotation matrix, the idea of the ICP algorithm is used to set the minimum rotation angle at each iteration. By continuously calculating the distance variance between the nearest points of the two sets of point clouds, the formula for calculating the variance is as follows: $D\left(x\right)=E({\left(x-E\left(x\right)\right)}^{2})$, where x is the distance from each point in the point cloud to the nearest point in the target point cloud. When the variance D(x) is minimized, the optimal rotation angle is obtained.

When calculating the translation matrix, this paper utilizes the cross-correlation coefficient of the two sets of two-dimensional point cloud data to calculate the translation distance in the x-axis direction of the point cloud to be registered. The average value of the y-values of the two sets of point cloud data is used to determine the translation distance in the y-axis direction. Cross-correlation is essentially the inner product operation of two arrays. In the linear space, it calculates the projection of the vector array of the point cloud to be registered onto the vector array of the target point cloud. The larger the inner product result is, the larger the projection, the smaller the angle between the two vectors, the more consistent the direction, and the higher the similarity between the two sets of point clouds. The formula for cross-correlation is as follows: (12)$$\left(x\right)*g\left(x\right)=\sum_{-\infty }^{\infty }f\left(x\right)g\left(x+\tau \right)d\tau$$
where f(x) and g(x) are the y-values of the control points on the centerline of the tunnel. The calculation process is shown in Figure 9. According to the centerline extraction algorithm proposed in this paper, the centerline control point arrays obtained are of equal length. Multiply and sum the corresponding values of f(x) and g(x) to obtain a cross-correlation result A. Then move each value in array g(x) backward by one position and calculate the corresponding values by multiplication and summation. Repeat this process for all values in the array to obtain a cross-correlation result array V=A,B,C,…,D of length 2n−1. Based on the position of the maximum value in array *V*, which represents the most similar cross-correlation coefficient between the two sets of point cloud arrays, calculate the position d of the backward movement of array g(x), thereby obtaining the translation distance of the point cloud data to be registered in the x-axis direction.

### 3.3. Algorithm for Tunnel Deformation Detection

The overall purpose of deformation detection in tunnels is to identify areas with significant changes from a global perspective. Due to the non-perfect alignment between the data acquisition devices and the tunnel’s position during two different data collection instances as well as the potential environmental changes within the tunnel between these instances, the quality of the point cloud data collected at each time may not be consistent. Consequently, there are no clearly corresponding points between the two sets of point cloud data, making it impossible to identify precise deformation signals. The deformation detection algorithm proposed in this paper is based on a point-to-point approach, utilizing the KNN algorithm to search for the centroid of neighboring points around the point of interest. It calculates the distance between the point of interest and its corresponding point to detect the deformation of the tunnel.

#### 3.3.1. Deformation Detection for a Whole Tunnel

An essential step in deformation detection algorithms involves finding corresponding points between two sets of point clouds and calculating the distances between these corresponding points. As shown in Figure 10, the black represents the target point cloud, while the green represents the point cloud to be tested. Here, the red points represent a point in the point cloud to be tested, the gray points represent k nearest neighbor points around the red point, and the yellow point represents the centroid of the gray points. The deformation distance, marked in red, is the distance to the yellow point. The specific algorithm process is as follows: (1) Select a point, p(x,y,z), on the point cloud to be tested. (2) Utilize the KNN (K-nearest neighbor) algorithm to select the nearest points $k_{1},k_{2},...,k_{n}$ on the target point cloud. (3) Calculate the coordinates of point p’s corresponding point on the target point cloud: $p^{\prime }\left(x^{\prime },y^{\prime },z^{\prime }\right)=\left(\frac{1}{k}\sum_{i=1}^{k}x_{i},\frac{1}{k}\sum_{i=1}^{k}y_{i},\frac{1}{k}\sum_{i=1}^{k}z_{i}\right)$, where *k* is the number of nearest neighbor points selected in the target point cloud. (4) Calculate the distance between point p and its corresponding point in the target point cloud as follows: $∥pp^{\prime }∥=\sqrt{{\left(x_{p}-x_{p^{\prime }}\right)}^{2}+{\left(y_{p}-y_{p^{\prime }}\right)}^{2}+{\left(z_{p}-z_{p^{\prime }}\right)}^{2}}$, which represents the deformation of point *p*. The accuracy of this algorithm is dependent on the density of the point cloud dataset and the number of nearest neighbor points. When the point cloud resolution is low and the number of nearest neighbor points is small, the accuracy of the detection algorithm may be affected.

#### 3.3.2. Deformation Detection of Tunnel Sections

Due to the fact that the cross-sections of mine tunnels are not strictly rectangular, it is not feasible to directly fit a standard rectangular frame to calculate the deformation of the points to be detected. In this paper, the distance from the points to be detected to the fitted centerline is calculated. By comparing the distances from the points to be detected and the corresponding points in the target point cloud to the centerline, the deformation of the points to be detected is calculated. When calculating the distance, considering the stress on the strata and the deformation of the tunnel in coal mines, this paper uses the Manhattan distance to calculate the horizontal and vertical changes between corresponding points separately. As shown in Figure 11, the distance from each point on the tunnel cross-section to the centerline point is the Manhattan distance of that point to the centerline point. The enlarged part in the figure represents the distance difference between corresponding points Δd = (coordinates of the point to be measured − coordinates of the target point) = Δx+Δy, which represents the deformation of that point. When phenomena such as roof collapse, sidewall bulging, and floor heaving occur in the tunnel, Δd<0. At the same time, by using Δx and Δy, the horizontal and vertical deformations of the tunnel at this cross-section can be further analyzed.

## 4. Experiment and Discussion

To validate the feasibility of the proposed algorithm, this study utilized point cloud data from underground mine tunnels. The experimental site spans approximately 342 m in length, featuring a rectangular cross-section with dimensions of approximately 5.8 m in width and 2.86 m in height, resulting in a cross-sectional area of 16.59 square meters. The study employed the digital LiGrip H120 handheld laser scanning system to collect point cloud data, and this system is capable of capturing data in a 280°×360° direction. The parameters of the laser scanning system are detailed in Table 1. Using the same equipment and methodology, two sets of point cloud data were collected from the same tunnel with a three-month interval. With the assistance of LiFuser-BP version 1.5.1 and LiDAR360 Version 6.0, alongside the SLAM algorithm, the study efficiently obtained high-precision 3D point cloud data through scanning. The point cloud data collected during the experiment are stored in the PLY format, including spatial coordinates X, Y, and Z and the reflection intensity of the tunnel. The two sets of point cloud data from the tunnel span approximately 350 m in length, with approximately 32 million points in total. This study focused on a subsection of the entire tunnel, extending approximately 15 m in the direction of the tunnel with approximately 1.2 million points, which includes the tunnel’s chambers. Our experimental setup consists of a Lenovo R9000p laptop manufactured in Beijing, China, equipped with an AMD Ryzen 7 5800H 3.20 GHz CPU and NVIDIA GeForce RTX 3060 GPU.

### 4.1. Extraction and Analysis of Tunnel Centerline

The extraction algorithm of the tunnel centerline is mainly based on the slicing of the tunnel. After the tunnel is sliced, the top and bottom lines of the slice and the left and right wall lines are extracted based on the x-values of the slice, respectively, using one-dimensional Gaussian kernel density estimation and two-dimensional linear kernel density estimation. The z-value and y-value of the center point are calculated by averaging. The extraction of the top and bottom lines of the tunnel slice using one-dimensional Gaussian kernel density estimation has a certain ability to resist noise interference. For complex slice situations, accurate top and bottom lines can also be extracted. As shown in the Figure 12, there are cross-sectional diagrams and kernel density estimation diagrams at different slice positions. Figure 12a,b are slices at x = 91,661 m; Figure 12c,d are slices at x = 91,662 m; Figure 12e,f are slices at x = 91,663 m. The red and green lines in the figure represent the bottom and top fitting lines, respectively, which are calculated from the abscissa of the red peak and the abscissa of the green peak in the kernel density estimation graph. The window width of kernel density estimation has a significant impact on the fitting results of the top and bottom lines. As shown in Figure 13, when the window width is 0.01 m in Figure 13a,b, the fitting line of the tunnel slice top plate is affected by noise points, deviating from the actual top plate point cloud. When the window width is 0.5 m in Figure 13e,f, the fitting line of the tunnel slice bottom plate is greatly affected by point clouds such as drainage channels and transport machinery, also deviating from the actual point cloud. When the window width is 0.1 m in Figure 13c,d, both the top and bottom fitting lines of the slice can coincide well with the actual point cloud, so this paper chooses a window width of 0.1 m.

When extracting the left and right wall lines of the tunnel, this paper uses two-dimensional linear kernel density estimation mainly because of the central symmetry of the Gaussian kernel, which leads to strong gradient characteristics in the kernel density estimation graph. Using edge detection algorithms to extract lines from this density map results in significant errors. As shown in Figure 14, the red points in the figure represent the xoy plane top view of the tunnel slice point cloud, Figure 14a shows the two-dimensional Gaussian kernel density estimation, and Figure 14b shows the two-dimensional linear kernel density estimation. It can be seen that the linear kernel density estimation has a significant difference in point density estimation at the boundary, which is more conducive to the extraction of boundary lines by edge detection and straight line extraction algorithms.

The left and right wall lines of the tunnel extracted according to the NURBS algorithm are shown in Figure 15. The blue control points in the figure are the curve control points recalculated by the NURBS algorithm, and the black lines are the left and right wall lines fitted according to the control points. After calculating the midpoint position, obtaining the centerline control points of the tunnel point cloud, and using the NURBS algorithm, the tunnel centerline obtained is more consistent with the center point. As shown in Figure 16, the green curve is the centerline fitted by the 3rd-degree B-spline curve, and the blue line is the straight line fitted by the least squares method. Both expressions of the actual tunnel centerline have certain deviations.

### 4.2. Point Cloud Registration Analysis

In this paper, the method of aligning centerlines is used to register the two tunnels, and the effect after registration is shown in Figure 17. The green and yellow represent the point cloud 1 and the target point cloud 2 to be registered, respectively, and the blue represents the shape after registration of the point cloud to be registered. The three tunnel centerlines in the figure are the centerlines of the three sets of point cloud collections mentioned above. The centerlines of the registered point clouds and the centerlines of the target point clouds basically coincide. Using the method proposed in this paper, the process of centerline registration of tunnel point clouds is shown in Figure 18. Figure 18a shows the projection of the centerline on the xoz plane. The yellow point cloud rotates around the intersection of the two fitted lines to the position where it overlaps with the blue point cloud, represented by the green point cloud. Figure 18b shows the points projected on the xoy plane of the centerline. The blue and green represent the point clouds to be registered and the target point clouds, and the red represents the registered point cloud obtained by rotation and translation. At the same time, this paper compares the results for the ICP point cloud registration algorithm, obtaining the yellow point cloud. It can be seen in Figure 18 that the registered centerline obtained by the algorithm in this paper is closer to the target point cloud.

From an overall perspective, the registration of tunnel point clouds achieves good results. For subsequent tunnel deformation detection, this paper further analyzes the cross-sections of the tunnel. As shown in Figure 19, the paper selects the same cross-sectional point cloud from different point cloud data and compares the proposed registration algorithm with ICP and NDT algorithms. The algorithm in this paper has a higher score of overlap. As shown in Table 2, five tunnel slices are selected to calculate the fitness score of registration. The fitness score is the root mean square error between corresponding points of the two point clouds. The smaller the fitness score is, the higher the registration accuracy. The algorithm proposed in this paper is compared with the classic ICP and NDT registration methods, which achieves a better fitness score in this scenario.

### 4.3. Tunnel Deformation Analysis

The method proposed in this paper is utilized to conduct deformation analysis on the registered point cloud data of the two phases mentioned above. The overall deformation effects of the point cloud are illustrated in Figure 20, showing the deformation of the tunnel from both internal and external perspectives with different colors representing the deformation values. The average deformation value calculated is 0.008 m. Points with deformation values within ±0.008 m are depicted in green, indicating minor deformation. Points with negative deformation values are depicted in yellow and red, indicating protrusion towards the interior of the tunnel. Points with positive deformation values are depicted in purple and blue, indicating recession towards the exterior of the tunnel. This method can similarly detect the deformation of arbitrary tunnel cross-sections, as shown in Figure 21.

To emphasize the locations of significant deformations in the tunnel and facilitate the rapid detection of safety risks by managers, this study sets a safety deformation threshold. The portions exceeding this threshold are marked in red, while the remaining safe areas are marked in green, as shown in Figure 22. Analysis reveals significant deformations due to differences in the environmental conditions within the tunnel during the collection of two sets of point cloud data. During the first data collection phase, the tunnel was in a non-production interval, with no coal miners or transportation activities. In contrast, during the second data collection phase, the underground tunnel was in operation, with significant coal accumulation visible on the conveyor belt in the image. The red point cloud near the right tunnel wall in the figure has larger deformations. Compared with the point clouds to be measured and the target point clouds, it can be seen that the tunnel to be measured has undergone roof bulging deformation.

To verify the accuracy of the algorithm, this paper randomly selects a point cloud slice in the tunnel and conducts experiments using the KNN algorithm to find corresponding points in the target point cloud. By selecting different number of nearest neighbor points, the number of nearest neighbor points with the smallest amount of average change is selected. As shown in the Table 3, when k = 1, the average change is not the minimum value due to the point cloud with measurement error, when k = 2 and 3, the average change is consistent, but the standard deviation of the change is smaller when k = 3, and when k > 3, the average change shows an increasing trend, shown in this paper, the number of nearest neighbor points is selected as 3.

This paper selects 10 sections and calculates the average change amount, standard deviation of the change amount, mean value of the change amount in the horizontal direction and mean value of the change amount in the vertical direction of the point cloud at the section, as is shown in Table 4. The average change amount of the section is 0.0267 m, which meets the data acquisition accuracy requirements of LiDAR equipment. Horizontal deformation can directly reflect the deformation of the roadway such as bulge and rockslide on the wall. Vertical deformation can directly reflect the deformation of the roadway such as roof sinking, bulge on the floor, and dented floors.

To further validate the accuracy of the deformation detection algorithm, this study employed the cross-measurement method at the same location during two sessions of collecting three-dimensional point cloud data in the tunnel, obtaining a deformation value of 0.015 m for the tunnel roof. Using this measurement as the first method, which is a benchmark. Different deformation detection methods were compared for accuracy at the same cross-sectional position, as shown in the table. The second method [46,47] utilized the registered point cloud data of the tunnel from this study and employed the MDP deformation detection algorithm to obtain the average deformation value of the cross-section roof. The third method [18] also utilized registered tunnel point cloud data, converting the Cartesian coordinates of the point cloud into polar coordinates centered on the cross-section midpoint, altering the corresponding point pairs, and calculating the average deformation value of the cross-section roof. The fourth method [36] involved fitting a straight line to the cross-section roof and calculating the average deformation value by comparing the distance of the point cloud to the line. As shown in Table 5, the deformation values obtained by the algorithm in this study are closest to those obtained by the cross-measurement method. Since the deformation of the roof often reflects vertical deformation, the algorithm in this study also demonstrates superior performance.

## 5. Conclusions

This paper proposes a coal mine roadway point cloud deformation detection algorithm based on mobile 3D laser scanning technology. The algorithm innovatively presents solutions in three aspects: roadway extraction for centerline identification, point cloud registration, and point cloud deformation detection. The algorithm achieves centimeter-level accuracy in deformation detection, consistent with the measurement precision of mobile 3D laser scanning technology. Compared with other mobile 3D scanning deformation detection algorithms, it demonstrates certain advantages. This algorithm can compute the centerline of the roadway in the presence of holes in the point cloud data and irregular shapes of roadway cross-sections, thus facilitating point cloud registration. Moreover, the algorithm does not require the deployment of special control points such as target balls or markers within the roadway, avoiding disruption to the environment inside coal mine roadways. It does not rely on known control point coordinates, hence possessing broader environmental applicability.

The deformation detection algorithm based on mobile 3D laser scanning technology proposed in this paper can complement stationary laser scanning or traditional contact-based deformation detection methods, forming a two-tier deformation detection approach for intelligent mines. Leveraging its low cost and high detection efficiency, the algorithm rapidly identifies overall deformation trends in underground roadways and local deformation quantities of roadway cross-sections. This facilitates the analysis of changes in roadway structural mechanics, providing reliable data support for roadway safety.

## Figures and Tables

**Figure 1 sensors-24-02299-f001:**
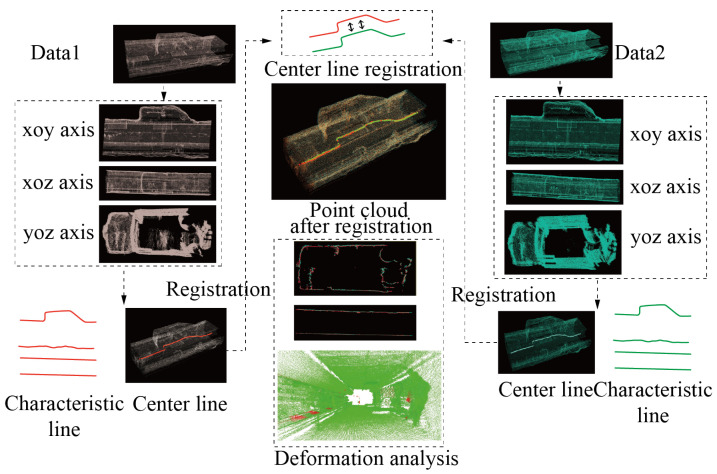
Algorithm for point cloud deformation detection in coal mine tunnels. The color lines show the tunnel boundary lines in the xoy and xoz planes.

**Figure 2 sensors-24-02299-f002:**
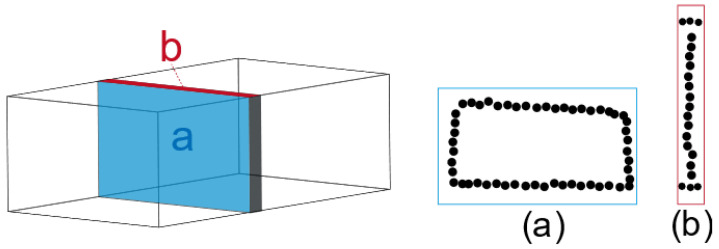
3D point cloud mapping 2D scatter plot. (**a**) Description of the scatter plot of the tunnel section. (**b**) Description of the horizontal projection scatter plot of the tunnel section.

**Figure 3 sensors-24-02299-f003:**
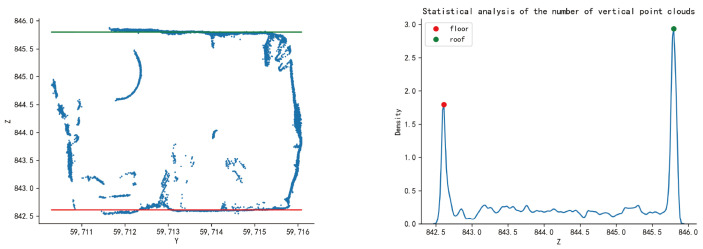
Extraction of the roof and floor lines of the tunnel based on the distribution density of the point cloud. Determine the location of the red and green lines in the left diagram based on the z-values of the red and green dots in the right diagram.

**Figure 4 sensors-24-02299-f004:**
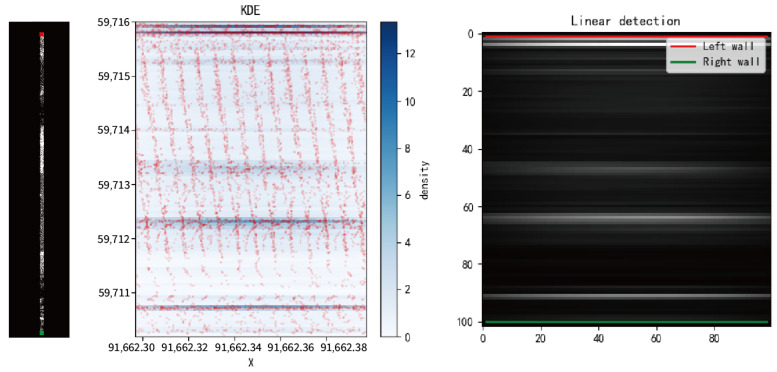
Extraction of the tunnel wall lines based on the distribution density of the point cloud.

**Figure 5 sensors-24-02299-f005:**
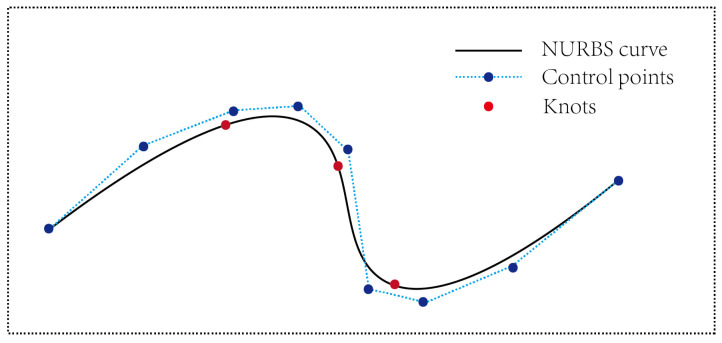
NURBS Fitting Curves.

**Figure 6 sensors-24-02299-f006:**
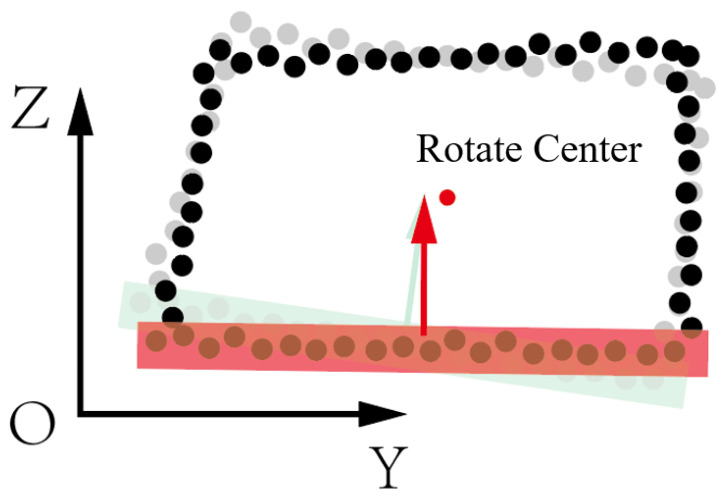
Rotate and translate in the zoy plane. The green points indicate original points. The red points indicate the points after rotation and translation. After rotation and translation, the floor of the tunnel is parallel to the horizontal.

**Figure 7 sensors-24-02299-f007:**
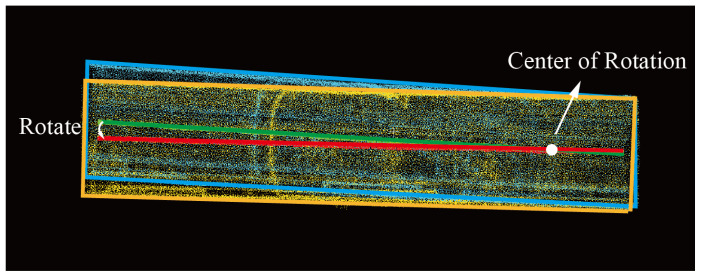
Alignment of the longitudinal section of a tunnel.

**Figure 8 sensors-24-02299-f008:**
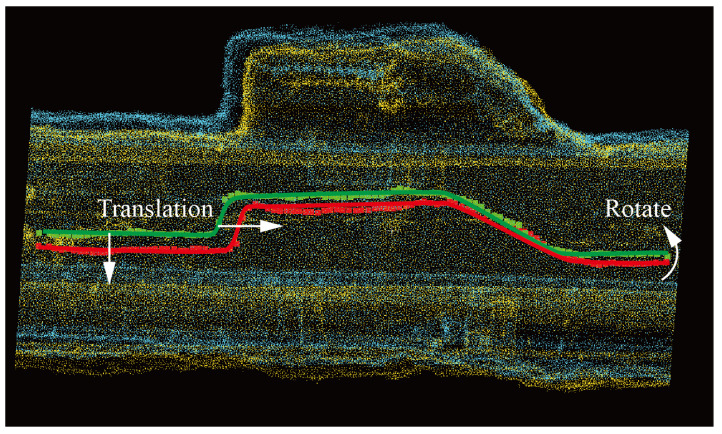
Alignment of tunnel planes.

**Figure 9 sensors-24-02299-f009:**
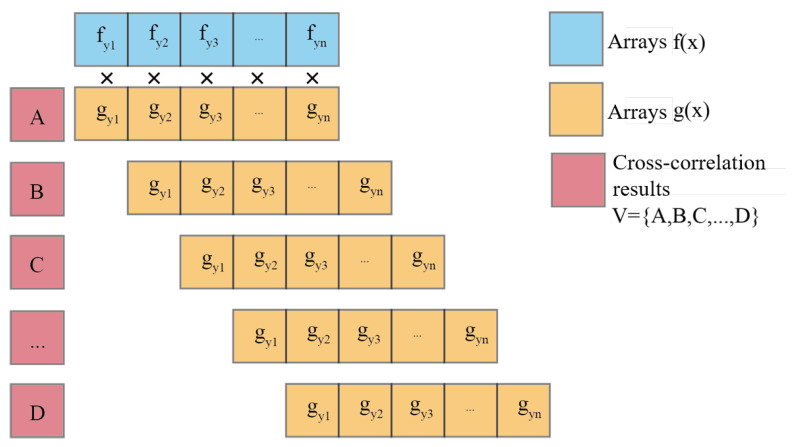
Principle of the Maximum Correlation Coefficient.

**Figure 10 sensors-24-02299-f010:**
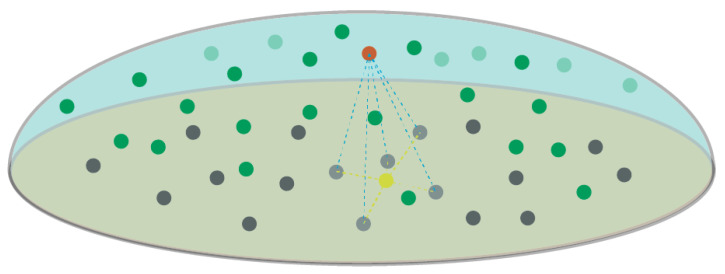
Calculate the corresponding point.

**Figure 11 sensors-24-02299-f011:**
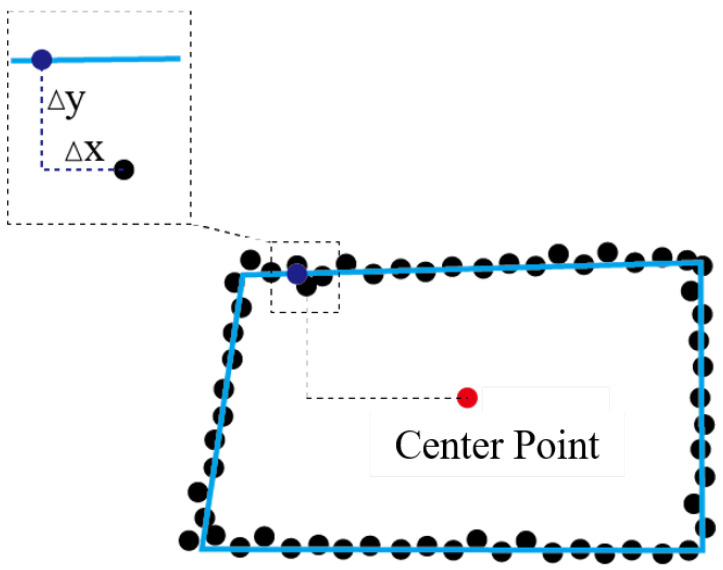
Calculate the distance between corresponding points.

**Figure 12 sensors-24-02299-f012:**
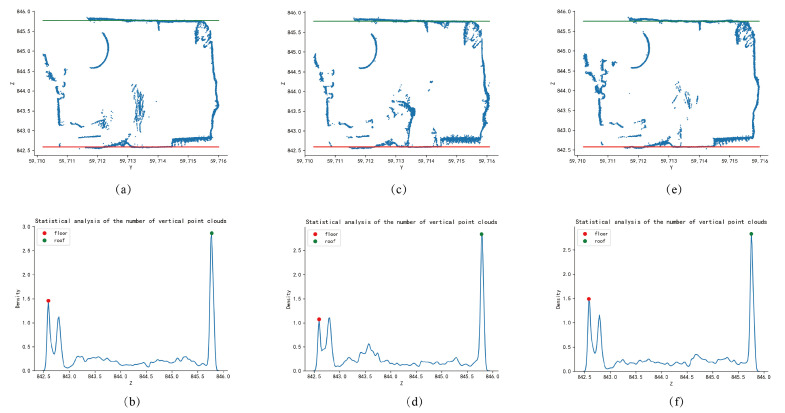
Cross-sections and 1D kernel density estimates at different slices of the tunnel point cloud. (**a**,**c**,**e**) are three different sections and the extracted top and bottom lines. (**b**,**d**,**f**) are the point cloud distributions corresponding to the three sections used to extract the top and bottom plate lines.

**Figure 13 sensors-24-02299-f013:**
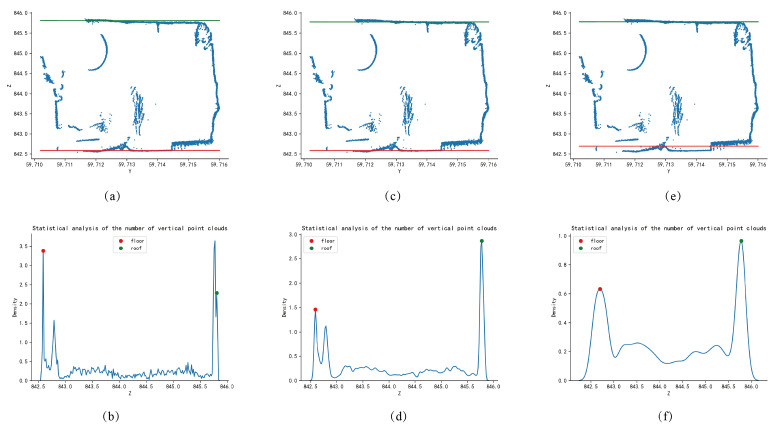
Effect of different window widths for kernel density estimation on roof and floor extraction. (**a**,**c**,**e**) are three different sections and the extracted top and bottom lines. (**b**,**d**,**f**) are the point cloud distributions corresponding to the three sections used to extract the top and bottom plate lines.

**Figure 14 sensors-24-02299-f014:**
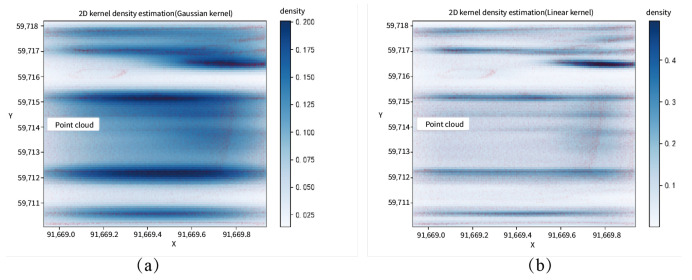
Comparison of two-dimensional Gaussian kernel density estimation and two-dimensional linear kernel density estimation.

**Figure 15 sensors-24-02299-f015:**
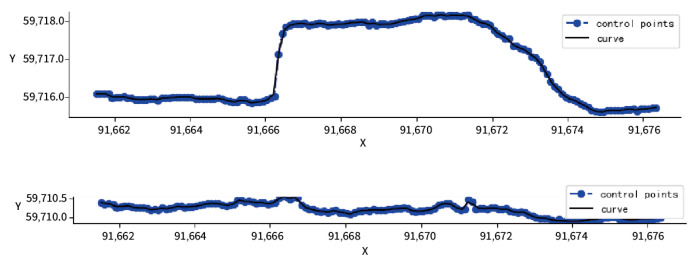
NURBS-based algorithm for fitting the left and right boundary lines of the tunnel.

**Figure 16 sensors-24-02299-f016:**
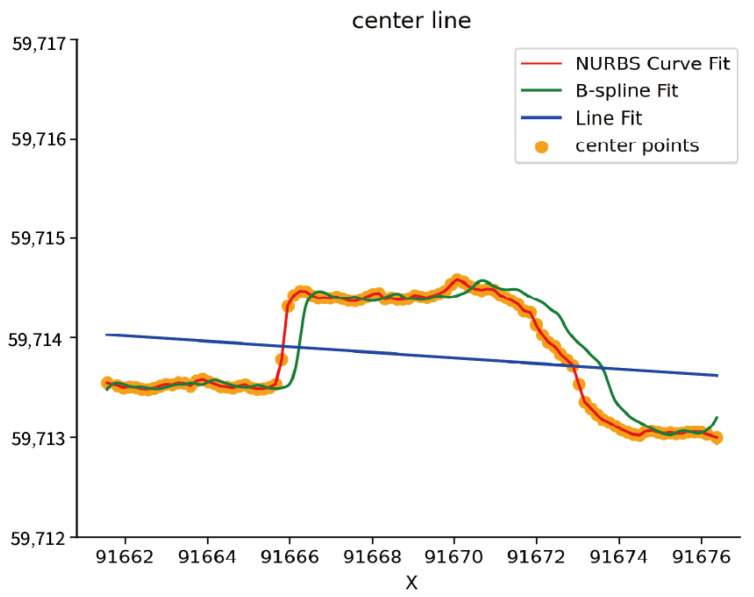
Comparison of Centerline Fitting with Different Methods.

**Figure 17 sensors-24-02299-f017:**
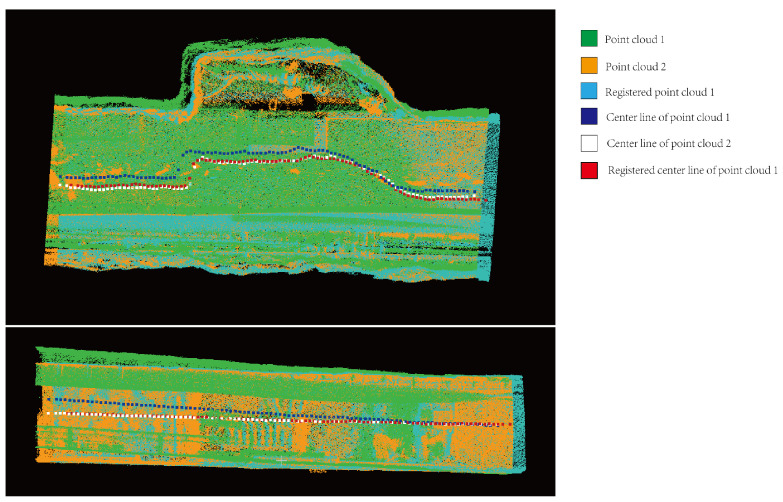
Tunnel point cloud alignment based on the tunnel centerline.

**Figure 18 sensors-24-02299-f018:**
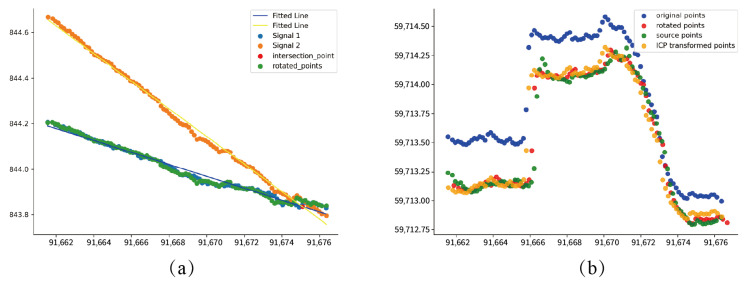
Alignment of the centerline of the tunnel. (**a**) shows the projection of the centerline on the xoz plane. (**b**) shows the projection of the centerline on the xoy plane.

**Figure 19 sensors-24-02299-f019:**
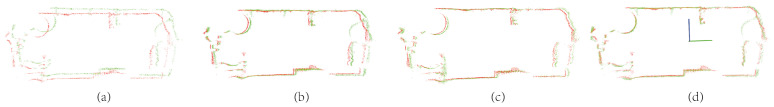
Comparison of point cloud alignment algorithms. The different color show the same slice in different point cloud. (**a**) shows the two tunnel cross-sections to be registered, (**b**) shows the two tunnel cross-sections registered by the ICP algorithm, (**c**) shows the tunnel cross-section registered by the NDT algorithm, and (**d**) shows the two tunnel cross-sections registered by the algorithm proposed in this paper.

**Figure 20 sensors-24-02299-f020:**
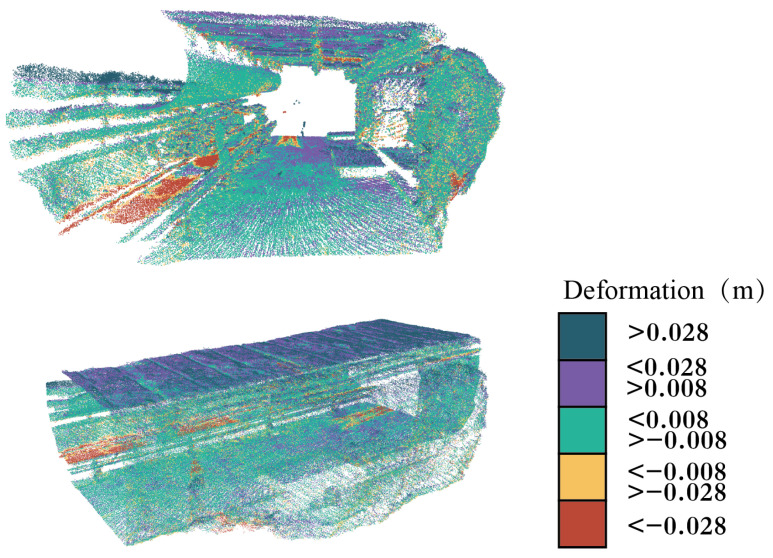
Global deformation analysis of the tunnel.

**Figure 21 sensors-24-02299-f021:**
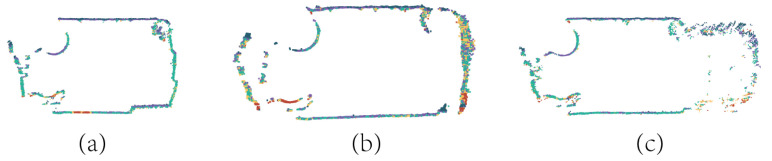
Section deformation analysis of the tunnel. (**a**–**c**) show the deformation variables of different cross-sections in a tunnel.

**Figure 22 sensors-24-02299-f022:**
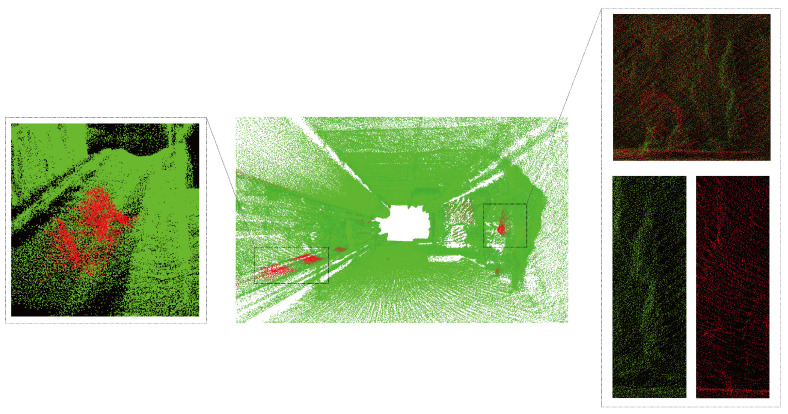
Key deformation detection analysis of the coal mine tunnel.

**Table 1 sensors-24-02299-t001:** LiGrip H120 Lidar system parameters.

Parameters	Data
Handheld size	L204 mm × W130 mm × H385 mm
Handheld weight	1.74 kg
Scanning field of view	280° × 360°
LiDAR accuracy	±3 cm
Point frequency	320,000 pts/s

**Table 2 sensors-24-02299-t002:** Alignment accuracy for different sliced point clouds.

Tunnel Slice	NDT (m)	ICP (m)	Ours (m)
1	0.0113	0.0091	0.0035
2	0.0087	0.0094	0.0028
3	0.0094	0.0084	0.0036
4	0.0126	0.0090	0.0038
5	0.0097	0.0086	0.0042

**Table 3 sensors-24-02299-t003:** Alignment accuracy for different sliced point clouds.

Number of Nearest Neighbor Points	Average Variation (m)	Standard Deviation of the Variation (m)
1	0.0314	0.0417
2	0.0305	0.0427
3	0.0305	0.0424
4	0.0308	0.0435
5	0.0309	0.0436

**Table 4 sensors-24-02299-t004:** Accuracy analysis of the deformation of the tunnel at different slices.

Slice X Position	Average Variation (m)	Standard Deviation of Variation (m)	Average Variation in Horizontal Direction (m)	Average Variation in Vertical Direction (m)
91664	0.0265	0.0188	0.0043	0.0091
91665	0.0244	0.0204	0.0034	0.0075
91666	0.0296	0.0245	0.0011	0.0108
91667	0.0242	0.0186	0.0002	0.0091
91668	0.0237	0.0191	0.0004	0.0083
91669	0.0255	0.0222	0.0004	0.0083
91670	0.0273	0.0338	0.0010	0.0086
91671	0.0301	0.0307	0.0007	0.0092
91672	0.0255	0.0322	0.0022	0.0089
91673	0.0305	0.0427	0.0031	0.0048
Mean	0.0267	0.0263	0.0017	0.0085

**Table 5 sensors-24-02299-t005:** Comparison of the accuracy of different deformation detection algorithms.

No.	Deformation Analysis Method	Deformation (m)	Deformation in Horizontal Direction (m)	Error (m)
1	Cross-sectional	0.0150	0.0150	0.0000
2	MDP	0.0227	0.0207	0.0077
3	Polar-coordinate	0.0364	0.0371	0.0214
4	Contour-fitting	0.0578	0.0623	0.0428
5	Ours	0.0209	0.0187	0.0590

## Data Availability

The data involved in the experiment are not publicly available due to the privacy issues of coal mining enterprises.

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
