# Peer review of "A Coal Mine Tunnel Deformation Detection Method Using Point Cloud Data"

_sensors, 2024, doi:10.3390/s24072299_

Round 1
Reviewer 1 Report (Previous Reviewer 1)
Comments and Suggestions for Authors
Dear Authors
The article has been significantly improved compared to the previous version and I am mostly satisfied with the corrections introduced. In particular, the additions regarding the literature review, the description of the proposed solution, as well as the analysis of the results obtained, in my opinion, increase the value of this article.
There is still no verification of the results obtained based on independent reference data with higher accuracy. However, the method of mutual alignment of trajectories itself is interesting and at this stage of research it can be used as a tool for identifying local changes between two measurement epochs and as such it is suitable for publication.
However, in future research, I suggest carrying out reference measurements in order to determine the accuracy parameters of the developed solution, both local and global, for the entire analyzed measurement trajectory.
I wish you success in your further research.
Author Response
We are very grateful to Reviewer for taking the time to review the paper so carefully. We have gained a lot. In the future, we will continue our research based on your suggestions. For more replies, please see the attachment.
Kind regards

Reviewer 2 Report (New Reviewer)
Comments and Suggestions for Authors
A method for coal mine roadway point cloud deformation detection is presented. The authors have combined some matured algorithms to register point clouds based on centerline extracted in the scans. The designer of the method can be improved with more illustrations. The method is interesting, however lack of new. Please, let us know what is the metric (mm, cm or m) obtained by NDT, ICP and Ours (see table 2). I supposed that the achieved global deformation has been influenced by the registration error task. I suggest include the metrics (mm, cm or m) in all tables, when mandatory. Why the deformation in vertical direction is higher than the deformation in horizontal direction? The conclusions are too obvious. We must give the reader every opportunity to evaluate the relative strenghts and weaknesses of various approaches depending on their research objective. Therefore, I would like to suggest that the authors add different ways of how improving the flaws for a particular method that still need to be explored. This can be added in the manuscript as future work and/or recommendations.
Then I hope that my comments in this review hopefully lead to a better paper, and should not be interpreted to be criticisms of the work. I wish the best success in making the revision and in your future efforts to take the laboriousness out of the processing of your interesting method.
Comments on the Quality of English Language
It is ok.
Author Response
We are very grateful to Reviewer for taking the time to review the paper so carefully. We have gained a lot. In the future, we will continue our research based on your suggestions. For more replies, please see the attachment.
Kind regards

This manuscript is a resubmission of an earlier submission. The following is a list of the peer review reports and author responses from that submission.
Round 1
Reviewer 1 Report
Comments and Suggestions for Authors
Interesting article, the authors present research results on the use of hand-held mobile laser scanners in examining the deformation of mining tunnels.
The article does not contain a sufficient review of the literature to present the essence of the issue, it is obvious that due to the popularity of this topic and the number of available publications, it is a subjective choice of the authors, however, the authors did not refer in any way to the accuracy requirements applicable to measurements of deformation of tunnels in mines.
The assumptions of the algorithm are presented clearly, step by step. The descriptions and procedure diagrams are sufficient to understand the assumptions and course of the experiments performed.
However, I have some serious comments about the article:
1. The authors declare, in accordance with the title and introduction, that the research aims to develop: "A deformation detection method for point cloud in coal mine tunnels", however, they nowhere refer to the accuracy requirements for carrying out such measurements, so it is impossible to verify whether the proposed method and selected measurement equipment are able to ensure the required accuracy. Even the use of a laser scanner with a measurement accuracy of +/- 3 cm raises doubts regarding its use for examining surface deformations in terms of the safety of using mining tunnels.
2. The use of hand-held mobile scanners for which trajectory alignment is based on SLAM algorithms in tunnel conditions should be particularly closely monitored and controlled, especially since in this case it is not possible to support trajectory alignment with GNSS technology. Relying only on data from inertial units, even those supported by Lidar Odometry or Visual Odometry algorithms, leads to increasing path length errors as well as direction errors. In the introduction, the authors refer to the efficiency of the proposed method in terms of the length of the measured tunnels compared to stationary scanning. However, the experiment was performed on a very short section of the tunnel (350 m), where the influence of trajectory alignment errors is small. This aspect of the problem was not discussed at all in the article.
3. I also have serious comments about the study in terms of accuracy in general. The proposed method should be verified based on reference data with known accuracy parameters, obtained either as a result of classical measurements or, for example, with a stationary laser scanner. In the presented article there is no accuracy analysis at all. Presented parameters of alignment of point clouds originating from the same measurement technology and alignment algorithm do not meet the condition of data control. They only confirm the repeatability of the results obtained in terms of the accuracy of the proposed technology.
4. Additionally, I have a few minor comments:
- some figures are difficult to read (e.g. 3, 10, 11, etc.), and some charts lack the description of the coordinate axes;
- it was assumed that the tunnel floor is horizontal line 381-382 and fig. 6: this assumption should be justified;
- some numerical parameters are missing units in the text, e.g. line 548, 553, 555
- the results of the ICP and NDT algorithms presented in Fig. 17 seem inadequate to the capabilities of these algorithms and may result from incorrectly adopted calculation parameters.
Moreover, the statement “The average amount of change in the horizontal direction of the tunnel is smaller than the average amount of change in the vertical direction, which indicates that the deformation generated by the top plate of the tunnel coming under pressure has a greater impact on the tunnel." is unjustified in terms of the accuracy of the tested hardware solution. Shown in table 4 values are below the measurement noise value, and the differences between the horizontal and vertical directions may result from the measurement characteristics of the device itself, infrastructure elements located on the walls and ceiling, etc.
This statement should be verified and confirmed by another, more precise measurement method.
Reviewer 2 Report
Comments and Suggestions for Authors
The paper proposed a method to automatically detect the deformation of coal mines by analysing point clouds acquired with a mobile 3D laser scanner.
The paper cannot be accepted in the current configuration because it does not meet Sensors' quality standards; some indications are given below.
3. Method
The authors, on the one hand, describe the method in an unclear and non-reproducible manner, while on the other hand, they verbose known algorithms (Canny edge detection in section 3.1.3; NURBS approximation in section 3.1.4; RANSAC plane fitting in section 3.2.2; least square fitting in section 3.2.3). For example, one thing that needs to be clarified is how the method can determine the centreline from the direction obtained with the PCA without an iterative optimisation method. How are the turns considered in the tunnel trajectory detection?
4 Experiment and discussion
This section is incomplete as only two point clouds from the same tunnel are considered, and the comparison with the state of the art is only made on the registration of the point clouds using the ICP and NDT algorithms, whose limitations are known in the literature. As a test case, the authors should have taken a coal mine monitored using a higher precision method (with stationary laser scanner), acquired with the mobile scanner, and verified that the proposed method introduced errors of an order of magnitude lower than required measures.
5 Conclusions
Conclusions, being without punctuation, are illegible.
The presentation of the method and the results described do not allow, in the current configuration of the paper, to verify the innovative aspects listed at the end of the introduction.
Comments on the Quality of English LanguageThe paper requires an improvement in the English language.